# Comparative Analyses of *Pediococcus pentosaceus* Strains Isolated from Milk Cattle Reveal New Insights for Screening Food-Protective Cultures

**DOI:** 10.3390/microorganisms13102244

**Published:** 2025-09-25

**Authors:** Sebastian W. Fischer, Nadine Mariani Corea, Anna Euler, Leonie Bertels, Fritz Titgemeyer

**Affiliations:** 1Institute for Hygiene and Public Health, University Clinics Bonn, 53127 Bonn, Germany; 2Department of Food Nutrition Facilities, FH Muenster, Correnstr. 25, 48149 Münster, Germany

**Keywords:** lactic acid bacteria, *Pediococcus*, antibacterial profile, food safety, bacteriocins, genome plasticity, genome fluidity

## Abstract

*Pediococcus pentosaceus* is a lactic acid bacterium used inter alia for the fermentation of milk, meat, vegetables, fruits, and even for brewing beer. Several health-promoting effects, such as antibacterial and antifungal activities or microphage and immune system stimulation, have been attributed. Apart from refining foods during the fermentation process, *P. pentosaceus* strains are added to meat and meat products as protective cultures to improve food safety, while leaving the organoleptic properties untouched. Since knowledge on the latter issue is still limited, we investigated 32 isolates from milk samples and teat canal biofilms regarding their antibacterial efficacy as a prerequisite for possible application as protective cultures. *P. pentosaceus* strains were unequivocally identified by DNA sequencing of the *rrnA* gene encoding 16S rRNA. Binary matrices obtained from random amplification of polymorphic DNA experiments showed that all isolates differed by more than 5% and thus represented subspecies. The antibacterial profiles against eight food-borne pathogens and food spoilage bacteria were determined. They efficiently combatted, although to various extents, Gram-negative bacteria such as *Pseudomonas aeruginosa* or *Salmonella enterica*, and Gram-positive bacteria such as *Staphylococcus aureus* and *Listeria monocytogenes*. Interestingly, acid production was dependent on the presence of the challenged pathogen and did not correlate with the extent of inhibition. Bioinformatic analyses of the genomes of the three top-ranked isolates revealed a pronounced genomic plasticity with a core genome of 1460 genes and additional 91, 130, and 161 unique genes, respectively. Each strain included a set of three, five, or six plasmids and was equipped with different genes encoding bacteriocins. The data suggest that multiple strains of *P. pentosaceus* should be included in order to optimize the selection of a culture for food preservation. The approach could also be applicable to other bacterial species.

## 1. Introduction

Microorganisms have been used for thousands of years to produce fermented beverages and food, which are very tasty and resistant to food spoilage [1,2,3,4]. In the early 1900s, Metchnikoff postulated that the astonishing average life expectancy of 87 years among Bulgarian farmers was due to the consumption of fermented milk products, such as those produced by *Lactobacillus bulgaricus* [5,6,7]. Since then, lactic acid bacteria (LAB) have been thoroughly investigated regarding their probiotic efficacies [4,5,8]. The result is a huge market that offers customers probiotic foods, nutraceuticals, and care products for human and animal health [4,9,10].

From a taxonomic point of view, LAB are grouped in the order *Lactobacillales* which includes the families Aerococcaceae, Carnobacteriaceae, Enterococcaceae, Lactobacillaceae, and Streptococcaceae [11]. They were recently re-classified based on a molecular phylogenetic approach at whole genome level, giving a total of 261 described species [12].

Members of the Lactobacillaceae, especially those from the genus *Lactobacillus* are most commonly used for food fermentation and protection [3]. Other family members such as the ten species of the genus *Pediococcus* are increasingly attracting attention in food fermentation, as probiotics and producers of small functional molecules, so-called postbiotics, and even for their skin-promoting effects [13,14,15,16]. Strains of *Pediococcus pentosaceus* are used in food production through for the fermentation of milk, vegetables, meat, fish, and fruit [17,18,19]. Health-promoting effects have been associated with macrophage and immune system stimulation, anti-inflammation, anti-cancer, treatment of infant colic, and fatty liver disease [6,18,20]. Furthermore, *P. pentosaceus* can combat pathogenic bacteria and fungi by secretion of lactic acid, hydrogen peroxide, diacetyl, diverse bacteriocins, and small peptides described as antibacterial and antifungal bacteriocin-like inhibitory substances (BLIS) [14,21,22]. Due to these antimicrobial properties, they are promising candidates for use in foods as protective cultures [18]. However, knowledge about the use of *P. pentosaceus* strains as food protectives is still very limited and there are only two relevant companies that sell *P. pentosaceus* strains as protective cultures for meat products [3].

To find novel functional LAB, we have chosen an environment where LAB are constantly exposed to a variety of pathogens. The udder of a cow meets this criterium perfectly since milk cattle suffer regularly from mastitis caused by a number of pathogens [23]. Foodborne pathogens are also found attached to the udder and are present in milk [24]. Among hundreds of isolates from foremilk and milk-biofilms from teat canals, we found that *P. pentosaceus* was one of the most predominant species. These were studied regarding their antimicrobial efficacies. We show that all isolates represent distinct subspecies and that each isolate exerts broad antibacterial activities against relevant pathogenic indicator bacteria. We further correlate this with the acidification characteristics of the isolates when exposed to respective indicator strains. We show that the profiles for antibacterial activity and acidification differ from isolate to isolate. To better understand the observed biodiversity, we analyzed the genomes of the three best-performing isolates regarding the genomic fluidity to corroborate the observed diversities. The implications towards the application of *P. pentosaceus* strains as protective cultures in foods are discussed.

## 2. Materials and Methods

### 2.1. Bacterial Strains, Growth Conditions, and Sample Collection

The *P. pentosaceus* strains of this study are listed in Table 1. They were cultivated on modified De Man–Rogosa–Sharpe (MRS) agar plates (Himedia, Mumbai, India) [25]. When appropriate, MRS plates were supplemented with 0.5 g/L cysteine (Carl Roth, Karlsruhe, Germany) (mMRS) to promote growth and with 10 mg/L bromophenol blue (Sigma-Aldrich, Darmstadt, Germany) (mMRS-bpb) to better distinguish colony morphologies with a blueish coloration [26]. Strains were freshly propagated in MRS broth for 48 h at 30 °C in an anaerobic jar. Eight bacterial species were employed as indicators and surrogates representing foodborne pathogens and food spoilage organisms (Table 2). These strains were cultivated in Luria–Bertani (LB) Broth (Carl Roth, Germany) after Lennox at 37 °C for 24 h. Brain heart infusion (BHI) broth (BD, Difco Fischer Scientific, Schwerte, Germany) was utilized for the cultivation of *L. monocytogenes*. Samples from teat canals were collected with swabs after teat disinfection. Swabs were introduced about 2 cm to obtain bacteria from the biofilm within the tip of the teat. Milk samples of 5 to 10 mL were subsequently withdrawn by milking to obtain bacteria present in the so-called foremilk.

### 2.2. Species Identification Through 16S rRNA Gene Sequencing

Genomic DNA extraction was performed by resuspending a single colony in 50 µL of 0.05 M NaOH. The mixture was incubated at 100 °C for 1 min. The lysate was precipitated by centrifugation. The supernatant with the genomic DNA was 1:10 diluted in sterile DNAse-free demineralized water. Polymerase chain reactions (PCR) were carried out in 25 µL, consisting of 12.5 µL oneTaq Polymerase Hot Start mastermix (New England Bio Labs, Frankfurt, Germany), 0.5 µL (10 pmol/μL) of primer F008 (5′-AGAGTTTGATCCTGGCTCAG-3′), 0.5 µL (10 pmol/μL) primer 535R (5′-TATTACCGCGGCTGCTGGCA-3′), 9 µL DNase, and nuclease-free water, and 2.5 µL (0.1 to 10 ng) DNA template. The resulting amplicon encompassed bases 8 to 535 of the gene encoding 16S rDNA representing the variable regions V1-V3. PCRs were carried out with an initial denaturation step at 95 °C for 10 min, followed by 40 cycles at 94 °C for 30 s, 60 °C for 30 s, 72 °C for 30 s. PCR amplicons were sequenced at Microsynth (Seqlab, Göttingen, Germany). The Basic Logical Alignment Tool (BLAST) service provided by the National Center for Biotechnology Information (NCBI, Bethesda, MD, USA) was used for species identification.

### 2.3. Random Amplified Polymorphic DNA Analysis (RAPD)

RAPD-PCR fingerprinting was carried out by applying the M13 minisatellite core primer (5′-GAGGGTGGCGGTTCT-3′) [27]. The PCR reaction mixture contained equal amounts of 10 ng genomic DNA, 2 µL primer (10 pmol/µL), and 10 µL of a OneTaq polymerase 2× Master Mix (New England Bio Labs, Frankfurt, Germany). The thermal cycling protocol involved an initial denaturation step at 94 °C for 7 min, followed by 40 cycles of denaturation at 94 °C for 30 s, annealing at 42 °C for 30 s, and extension at 68 °C for 4 min. A final extension cycle was performed at 68 °C for 10 min (modified after Sirichoat et al. [28]). PCR products were subjected to electrophoresis on a 2% agarose gel (AppliChem, Darmstadt, Germany).

### 2.4. Hydrogen Peroxide Production

Hydrogen peroxide production was conducted following the method described by Tomás et al. [29]. Fresh cultures of each isolate were plated on mMRS agar plates supplemented with 1 mM 3,3′,5,5′-Tetramethylbenzidin (TMB) and 2 U/mL of type II horseradish peroxidase (Carl Roth, Germany). Agar plates were incubated anaerobically for 48 h at 30 °C. Subsequently, the Petri dish lid was removed for exposing the colonies to fresh air for 10 min. Hydrogen peroxide production was visible when the yellowish colonies turned black. Data were reproduced in triplicate.

### 2.5. Antimicrobial Activity and Acidification

Antimicrobial activities of *P. pentosaceus* isolates against foodborne pathogens and food spoilage organisms were evaluated with lawn-on-spot assays modified as described (Table 2) [30,31]. Isolates were inoculated from a liquid overnight culture by positioning 2 µL into the center of an MRS agar plate. Agar plates were incubated under anaerobic conditions at 30 °C for 48 h. The freshly grown colonies were overlaid with 10 mL of LB-soft agar (0.8%) or BHI-soft agar (0.8%) in the case of *L. monocytogenes*. Before pouring, indicator strains were applied by adding 0.2 mL of a cell suspension of 1.0 × 10^7^ CFU/mL. Following a 20 to 30 h co-cultivation at 37 °C, the diameter of the zone of growth inhibition, from which the colony diameter was subtracted, was measured in millimeter. Acid secretion of the *P. pentosaceus* isolates was routinely determined with color fixed indicator sticks with a range between pH 3.6 to 6.1 (pH FIX strips; Carl Roth, Germany). In this way, it was possible to measure changes in pH by comparing the pH in the inhibition zone with the pH of the indicator strain grown on the same media in the absence of a *P. pentosaceus* isolate (control).

**Table 2 microorganisms-13-02244-t002:** List of indicator strains.

Species	Origin	Comment
*Bacillus subtilis*	laboratory strain collection	surrogate food spoilage
*Citrobacter koseri*	laboratory strain collection	fish-borne pathogen
*Listeria monocytogenes*	DSM 20600	food pathogen
*Pseudomonas aeruginosa*	ATCC 15442	food spoilage and pathogen
*Salmonella enterica* ^1^	LT-2	food pathogen
*Staphylococcus haemolyticus*	laboratory strain collection	surrogate food pathogen
*Staphylococcus warneri*	laboratory strain collection	surrogate food pathogen
*Staphylococcus aureus*	laboratory strain collection	food pathogen

^1^ re-classified from originally S. typhimurium [32] to S. enterica ssp. enterica ser. Typhimurium [33].

### 2.6. Data Analyses and Statistics

Data analyses comprised subspecies identification, antimicrobial activity in vivo, and the relationship between antimicrobial compounds and acid secretion. To do so, the statistical and computing package R (v4.2.1) and RStudio (v2023.03.0+386) with packages tidyverse (v2.0.0), readxl (v1.4.2), ggplot2 (v3.4.2), vegan (2.6-4), and ggdendroplot (v0.1.0) [34,35,36,37,38,39] were used. Subspecies identification was based on the analysis of the data derived from RAPD-PCR DNA band patterns. Therefore, the unweighted pair group method with arithmetic mean (UPGMA) was applied by using the hclust algorithm together with the Dice dissimilarity coefficient matrix [40,41] from the R base package. Phylogenetic trees were generated using a threshold of ≥5% dissimilarity to distinguish clonal isolates [40,42,43,44,45]. Strains were analyzed for antimicrobial activity in vivo via heatmapping and cluster analysis, by normalizing inhibition zone diameters across strains by z-score calculation. Hierarchical clustering of strains and indicators was performed using Euclidean distances and complete linkage in the hclust function, visualized through heatmap and dendrograms in ggplot2 and ggdendroplot. The above-mentioned statistical methods were also used to evaluate the data obtained from acidification experiments. Pearson correlation and scatterplot visualization performed on the raw replicate values assessed the contributions of antimicrobial compounds and acid secretion at a 95% confidence level.

### 2.7. Genome Analyses

The recently sequenced genomes of strains 05.5 8-1, 13.7 2A-1 and 13.7 13A-2 were retrieved as fasta files from BioProjekt ID PRJNA1297324. Gene annotations, chromosomal organization, and circular maps were obtained using the bioinformatic tools Bakta, Prokka, and FastANI as implemented into the Proksee server [46,47,48,49]. A Venn diagram showing the core genomes, the pan genomes, and singleton genes was computed with EDGAR 3.0 on the basis of the gene annotation list created with Bakta [46,50]. Genes encoding bacteriocins were screened by uploading fasta files of chromosomes and plasmid DNA sequences into the BAGEL4 server [51].

## 3. Results

### 3.1. Isolation of P. pentosaceus Strains from the Cattle Udder

During screening for novel food protective strains, we isolated several hundred microbial strains from the udder of milk cows from 15 farms located in Münsterland, Germany. Among those were 33 strains of *P. pentosaceus* as demonstrated by the complete DNA sequence identity of a 483 base pair fragment of the 16S rRNA-encoding gene with reference *P. pentosaceus* ATCC 25745. One isolate, that produced the antibacterial compound hydrogen peroxide, was excluded since such strains can alter the taste of food products by oxidation processes, an undesired characteristic for the use of protective cultures in foods [52,53].

### 3.2. Genetic Biodiversity

Random Amplified Polymorphic DNA (RAPD) analyses were applied to study the genetic diversity of the isolates. Therefore, chromosomal DNA samples of the 32 *P. pentosaceus* isolates were subjected to RAPD-PCR. Representative band patterns of 21 isolates from farms 4 and 13 are shown in Figure 1.

A matrix of the fragment size distribution was generated and used for UPGMA cluster analyses to transfer the gene fragment patterns into phylogenetic relationships (Figure 2). Across the four dairy farms, from which *P. pentosaceus* strains were obtained, the relative similarity ranged from 20% to 85%.

While the spatial proximity of isolates within the phylogenetic tree suggested a higher degree of genetic similarity and potential relatedness among individuals, none of the isolates within a farm exhibited a phylogenetic dissimilarity of less than 5%, the threshold to discriminate subspecies [44]. Hence, all isolates were classified as distinct subspecies within one farm.

### 3.3. Antimicrobial Profiles

To uncover the most important prerequisite for a protective culture, the antimicrobial efficacy, we took 13 diverse isolates and tested them against a set of indicator strains comprising food pathogenic, spoilage bacteria, and representative surrogates. Therefore, lawn-on-spot assays were conducted to observe growth inhibition by the formation of inhibition zones. A heatmap that was derived from calculated z-values, representing normalized inhibition zone sizes, is presented in Figure 3. Most susceptible were *Pseudomonas aeruginosa* and *Salmonella enterica* situated in the first subcluster on the left with inhibition zones between 34 and 56 mm.

A second cluster of indicators comprised the Gram-negative, enteric bacterium *Citrobacter koseri* and the three Gram-positive *staphylococci*, ranging from inhibition zone sizes of 17 to 24 mm. While *C. koseri* was generally more sensitive to *P. pentosaceus* isolates, three of the 13 isolates were more effective against the *Staphylococcus* strains. The antimicrobial effect towards the third cluster, consisting of the Gram-positives *L. monocytogenes* and *Bacillus subtilis*, was below average. It is noteworthy that *L. monocytogenes* was generally susceptible to most isolates, especially to isolate 13.7 8A-4. The zones of inhibition for *L. monocytogenes* ranged from 8 to 17 mm. The only indicator that showed very weak or, in three cases, even zero susceptibility was the food spoilage surrogate *B. subtilis*.

### 3.4. Impact of Acidification on Antibacterial Activity

After the antimicrobial profiles had been established, it was important to know which isolates had a low ability to acidify in order to avoid undesirable changes in the flavor of food. Figure 4 shows two representative agar plates from lawn-on-spot assays. The color of the pH strip on the left corresponds to pH 6.1, indicating weak acidification of the soft agar. By contrast, the pH strip on the right plate indicates acidification with a pH of 4.4. In this way, we investigated the extent of acidification of each *P. pentosaceus* strain against all indicator strains.

Acidification of the medium in response to the indicator strains was extremely variable. Notably, the presence of *S. enterica* induced the strongest acidification by an average decrease in pH of 0.79 (±0.1) units. By contrast, *C. koseri* yielded the mildest acidification with a pH drop ranging around 0.1 (±0.08) units. All isolates behaved quite similarly in acidification with an average pH reduction of 0.26 (±0.04) units (Figure 5).

To elucidate the influence of acid secretion on the measured antimicrobial activity, a Pearson correlation analysis was performed, and non-linear correlations were visualized through scatterplot analysis. Figure 6 shows that acid secretion had a significant effect on the measured antimicrobial activity for *S. aureus* and *L. monocytogenes*, with a correlation coefficient of −0.600 and a *p*-value of 0.000 and −0.330 and a *p*-value of 0.042, respectively. No other indicator organism demonstrated sensitivity solely through acid secretion.

Out of the thirteen isolates, six revealed a significant moderate to strong correlation between acid secretion and measured antimicrobial activity across all indicators. These were the isolates 02.3 3A-1, 13.7 10-2, 13.7 10A-6, 13.7 17-3, 13.7 9A-11, and 13.7 8A-4 (Figure 6B).

Based on the above presented results, *P. pentosaceus* strains were ranked by combining breadth of activity, aggregate potency, consistency of inhibition, and independence from acidification (Table 3). Breadth of activity reflects the count of indicator pathogens for which a strain’s inhibition z-score exceeded the overall mean. Aggregate potency refers to the sum of those positive z-scores, whereas consistency of inhibition denotes the average z-score across only the above-mean values. Finally, we excluded any strain whose inhibition profile remained significantly correlated with pH reduction, thereby focusing on candidates whose antimicrobial effect cannot be attributed solely to acid production.

Considering the above-mentioned criteria, strain 13.7 13A-2 emerged as the best performer overall. It inhibited five of eight tested pathogens at a level above the population mean and reached the highest total cumulative z-score (2.75), with a mean positive z-score of 0.344. Its Pearson test yielded *p* = 0.1935, confirming that its inhibition was not driven by acidification. A second tier of isolates—13.7 2A-1, 05.5 8-1, 22.4 8A-4, 13.7 22A-4, and 13.7 18A-3—inhibited four pathogens above average, with cumulative z-score values between 2.84 and 3.09 and non-significant correlation *p*-values (0.069–0.764). Finally, 29.4 19-1 inhibited three pathogens above mean level (sum = 2.81, mean = 0.352, *p* = 0.152).

This group of seven *P. pentosaceus* isolates were probably the best candidates to use as protective cultures in foods as they combined superior inhibition against most tested pathogens with mild acid production.

### 3.5. Genome Analyses

The observed variations in antibacterial activity and acidification were in accordance with RAPD data, since all isolates differed at the subspecies level as defined by Nielsen et al. [44].

To further corroborate these findings at a molecular level, we examined the previously published genomes of the three best candidates. The chromosomes differed in size, namely 1.77 Mb, 1.73 Mb, and 1.84 Mb for strains 13.7 13A-2, 05.5 8-1, and 13.7 2A-1, respectively, and accordingly in the number of predicted genes (Table 4). They further harbored an unequal set of three, five, and six plasmids (Table 4). Overall, they exhibit the same gene order, as can be inferred from the five red-labeled gene loci that encode the paralogs of the 23S RNAs (Figure 7A–C). Another measure was comparing the relative locations of homologous genes. Therefore, we determined the distance from base one of the circular maps, the first nucleotide of the *dnaA* gene that is positioned at 12 o’clock, to three conserved genes (*uvrA*, *polC*, *secY*) located at about three o’clock, six o’clock, and nine o’clock. Their distances varied from about 9 to 94 kb, an indication for genetic arrangements caused by horizontal gene transfer or the activity of transposable elements (Table 4).

This became more obvious by looking at the black graphs displaying the G + C content. Figure 7D shows a magnification of the three chromosomal regions around the first 23S rRNA gene locus. The course of the G + C content of the left graph shows two regions with significantly lower G + C content within the genome region of strain 13.7 2A-1, while the other two have one or none of these regions, respectively.

The distribution of genetic variation between the genomes was then addressed by analyzing the Average Nucleotide Identity (ANI) and the pan genome (Figure 8). The FastANI graph of Figure 8A shows that the genomes shared 98 to 99% sequence identity at the nucleotide level and that the first half of the chromosomes exhibits higher plasticity. The Venn diagram in Figure 8B shows a core genome of 1460 genes. The strains differ in 91, 130, 161 unique genes so called singletons. In conclusion, the molecular data confirm the genetic fluidity among *P. pediococcus* strains.

### 3.6. Bacteriocin-Encoding Genes

Bacteriocins are important peptides with diverse antibacterial activities. Hence, the chromosomal and plasmidal DNA sequences were screened for the presence of bacteriocin-encoding genes using the BAGEL4-server, which, to our knowledge, emphasizes the most comprehensive database [51]. As shown in Figure 9, gene regions were detected within the chromosomes of 13.7 13A-2 and 13.7 2A-1 that harbor a putative *penA* gene for the bacteriocin penocin A (light green). Directly upstream on the left is an associated gene that might confer immunity to Penocin A (red). Two open reading frames are situated downstream, encoding a putative two-component system consisting of a histidine kinase and a response regulator gene. The four genes upstream encode a putative PEP-dependent phosphotransferase system (PTS) for the consumption of ß-glucosides and a gene encoding a IIA^Glc^ homolog of the PTS, which in many bacteria functions in glucose transport and down-regulation of second choice carbon sources when glucose is available [54]. A *penA* gene locus was not found in the genome of strain 05.5 8-1, which instead harbors a gene for enterolysin A.

Genes for the closely related Pediocin A were found together with bacteriocin immunity genes on plasmids from 13.7 2A-1 and 05.5 8-1. Pediocin A and the adjacent immunity proteins were highly conserved sharing 95% and 98% amino acid identities, respectively. The surrounding genes, including some for bacteriocin export (pink), were quite dissimilar. When all four bacteriocin proteins were aligned, they exhibited an overall protein sequence identity of 45%, all harboring the conserved motif “YGNGV(L)” and two conserved cysteins that form disulfite-bridges to determine the three-dimensional structures of these class II bacteriocins.

## 4. Discussion

*P. pentosaceus* strains isolated from a natural ecosystem, the udder of milk cows, exhibit a pronounced biodiversity. All strains were able to inhibit a range of foodborne pathogens. However, antibacterial efficacies and acid production were surprisingly variable. The analysis of three genomes of the best-performing strains revealed a multifaceted genetic plasticity in genome size, plasmid numbers, genes encoding bacteriocins, and, in particular, unique genes.

The data obtained from the RAPD experiments showed that all isolates differed genetically by more than 5%, a threshold defined by Nielsen et al. for distinguishing subspecies [44]. Comparable results, although in less detail, were reported from plant-derived *pediococci* that were isolated from traditional Ethiopian foods such as tef dough and kocho [55]. Quite similar observations were found when *L. plantarum* strains were compared [56]. In a comprehensive work, Rossetti and Giraffa analyzed more than a thousand LAB strains from raw cheese [27]. They established a databank of RAPD-profiles which can be used to identify an unknown isolate just by the gene fragment pattern. Since *P. pentosaceus* strains were not included, the data presented in this paper add to this. To our knowledge, they are the first of a series of *P. pentosaceus* strains from a particular environment.

All isolates exhibited antibacterial properties against the examined range of food-borne pathogens. Surprisingly, the obtained profiles were distinctly different. On a ranking that was based on activity breadths, significance of inhibition combined with low acidification showed that *P. pentosaceus* 13.7 13A-2 was the most efficient one. It was able to inhibit five out of eight pathogens above average. The second-best performing isolates were a group of five that inhibited four out of eight above average. The usage of the antimicrobial potential of *P. pentosaceus* in foods has been addressed recently on one *P. pentosaceus* and one *P. acidilactici* isolate from silage [57]. It was reported that they could combat 74 *L. monocytogenes* isolates and 27 different *enterococci* that were resistant to vancomycin. Furthermore, it was shown that these isolates inhibited many fungal species, most likely by several antifungal metabolites [57]. In another study, the addition of a mixture of *P. pentosaceus, P. acidilactici*, and *L. plantarum* to alfalfa sprouts reduced the titers of *L. monocytogenes* by 4.5 log and of *S. enterica* by 1.0 log, respectively [58]. In accordance with our observation that isolates of the same species have different antimicrobial profiles, Santini et al. published that different strains of the same LAB species as well as *bifidobacteriae* exhibited different efficacies against three *Campylobacter* species [59]. 

When applied to a food product, the production of acid by a protective culture is an important aspect in the control of foodborne pathogens. However, acidification may change the organoleptic characteristics of the food product [60,61]. Our approach to measure the acidification capacity of each isolate against each pathogen allowed us to select weak acid producers while maintaining antimicrobial efficacy. It should be noted that acid production not only depended on the respective *P. pentosaceus* strain, but also on the presence of the competing strain. We would like to emphasize that such correlation studies on antibacterial efficacy and acid formation have not yet been investigated. It remains to be demonstrated if this observation accounts also for other LAB.

Genome analyses of the three best-performing strains, which we recently determined, corroborated the genetic fluidity and thus support the data obtained from the RAPD experiments. Analysis of the pan genome showed that 1460 genes were present in the core genome, while 91, 130, and 161 represented unique singletons [50]. These findings were confirmed by counter-checking further available *P. pentosaceus* genomes [62]. The present plasmids harbored genes for the metabolism of unusual carbon sources and for bacteriocins. These characteristics are very common in bacterial plasmids in order to facilitate gene transferbetween microorganisms [63].

Numerous reports focus on bacteriocins produced by *pediococci* [64,65]. These are usually small peptides with antibacterial activity [66]. In particular, the presence of pediocin and pediocin-like bacteriocins from *P. pentosaceus* in meat products has been thoroughly examined [67]. We detected chromosomal and plasmidal genes for the class II bacteriocins pediocin A and penocin A and class III enterolysin, together with genes for immunity and export. The corresponding genes, genetic organization, and functions are commonly found in pediococci from various environments [64,65,68,69]. It should be noted that they are not constantly expressed since this is influenced by environmental factors such as pH, temperature, quorum sensing, carbon sources, essential amino acids, vitamins, and the energy state of the cell [70,71,72].

## 5. Conclusions

This study aimed to characterize a group of *P. pentosaceus* strains regarding their potential to serve as protective cultures in foods. Therefore, biodiversity, antibacterial efficacies, as well as acid production were examined. The results showed that each isolate was able to inhibit foodborne pathogens. The extent of inhibition and the antibacterial profiles of each isolate were distinctly different. The same was observed for acid production which, interestingly, depended also on the presence of the respective pathogen. The data were correlated by describing genomic analyses of the three best strains, demonstrating a pronounced genetic plasticity. Based on these findings, we suggest that the approach for selecting the best *P. pentosaceus* protective culture for a food product should include several strains. The same blueprint may also apply to other bacterial species. However, we are aware that further tests beyond those presented here are necessary before a protective culture can be definitively applied. These will include challenge tests for each food product under real production conditions to clearly demonstrate that the respective pathogens are controlled by the protective culture.

## Figures and Tables

**Figure 1 microorganisms-13-02244-f001:**
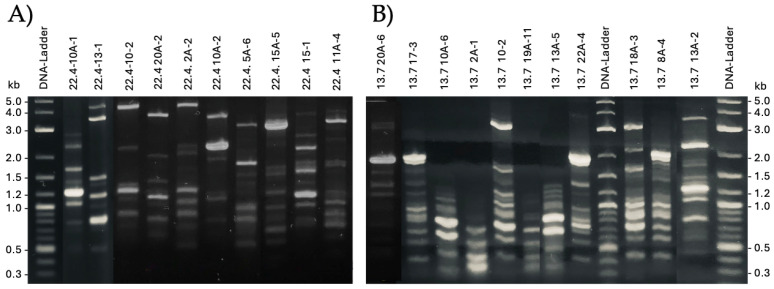
RAPD analyses. The depicted 2% agarose gels show the resulting gene fragment patterns. (**A**) Isolates from farm 4;isolates (**B**) isolates from farm 13.

**Figure 2 microorganisms-13-02244-f002:**
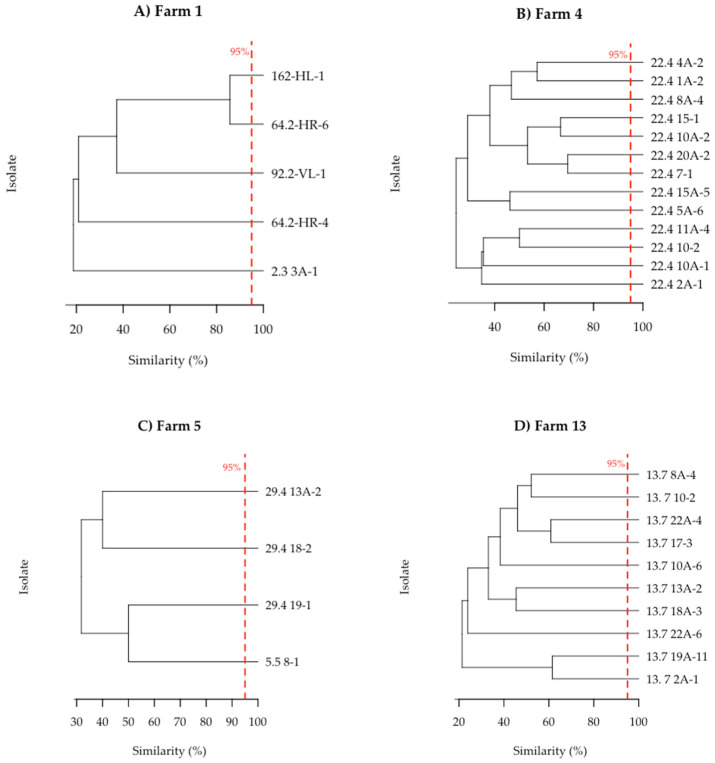
Depiction of the phylogenetic relationships within isolates from one farm. Relative phylogenetic distances are indicated by scale. The red line depicts the 95% similarity criterium to define a subspecies. (**A**) farm 1 (n = 5); (**B**) farm 4 (n = 16); (**C**) farm 5 (n = 5); (**D**) farm 13 (n = 11).

**Figure 3 microorganisms-13-02244-f003:**
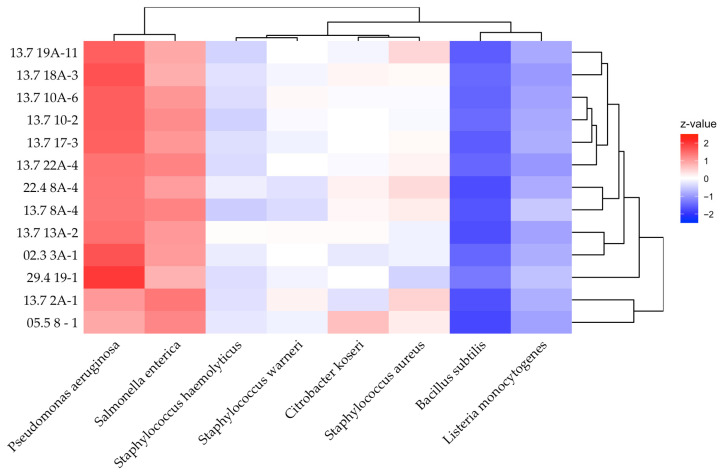
Heatmap of antibacterial activities. The color scale shows the normalized z-values. Indicators are ordered by decreasing susceptibility from left to right. Their clustering is shown above. The clustering on the right shows the subspecies of *P. pentosaceus* according to the similarity of their antibacterial profiles. Experiments were carried out in triplicate.

**Figure 4 microorganisms-13-02244-f004:**
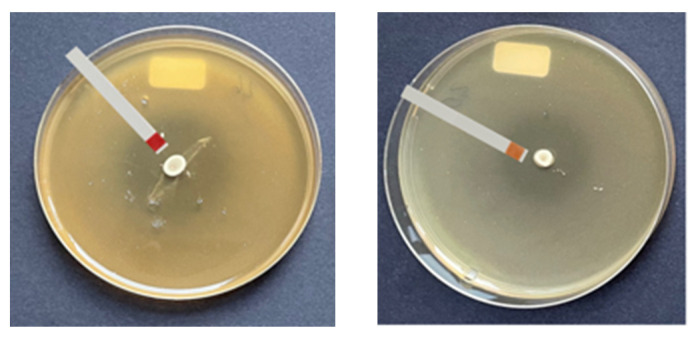
Measurement of acidification within the zones of inhibitions with pH strips employed in lawn-on-spot tests. **Left**: pH 6.1; **right**: pH 3.6.

**Figure 5 microorganisms-13-02244-f005:**
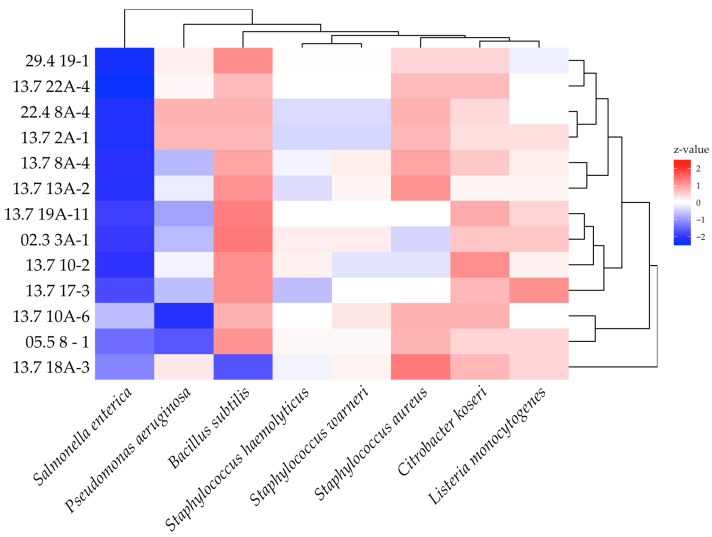
Heatmap of pH drop of *P. pentosaceus* isolates against indicator strains. The columns show the normalized z-values of acidification ranging from blue (strong) to white (fair) to red (low). The clustering of indicators is displayed on the top, while clustering of the isolates is shown on the right. Data represent the average of triplicates.

**Figure 6 microorganisms-13-02244-f006:**
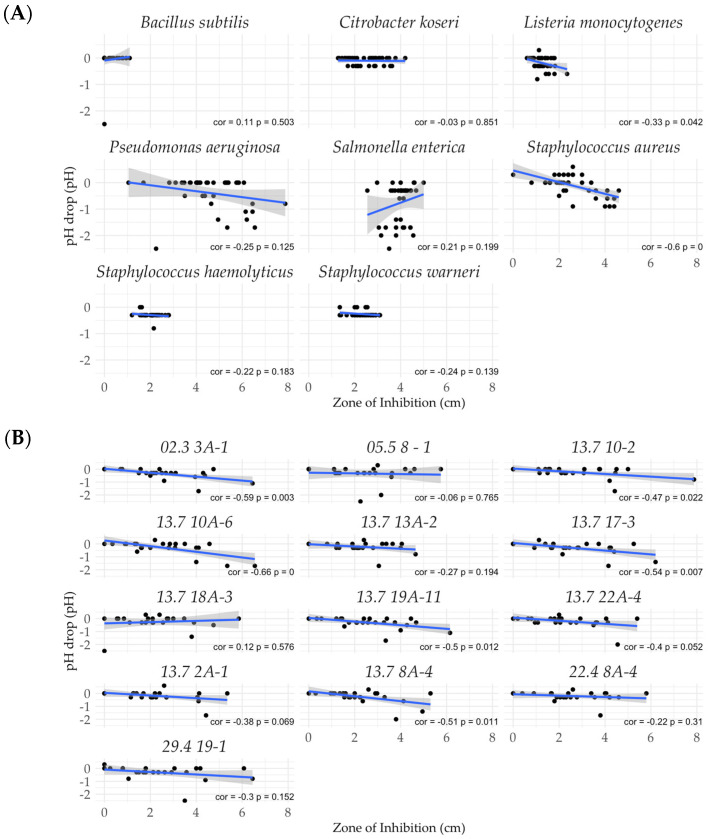
Pearson correlation analysis. The drop in pH versus the inhibition zone is shown for (**A**) indicators and (**B**) *P. pentosaceus* isolates. r, correlation coefficient; *p*, *p*-value significance of <0.05.

**Figure 7 microorganisms-13-02244-f007:**
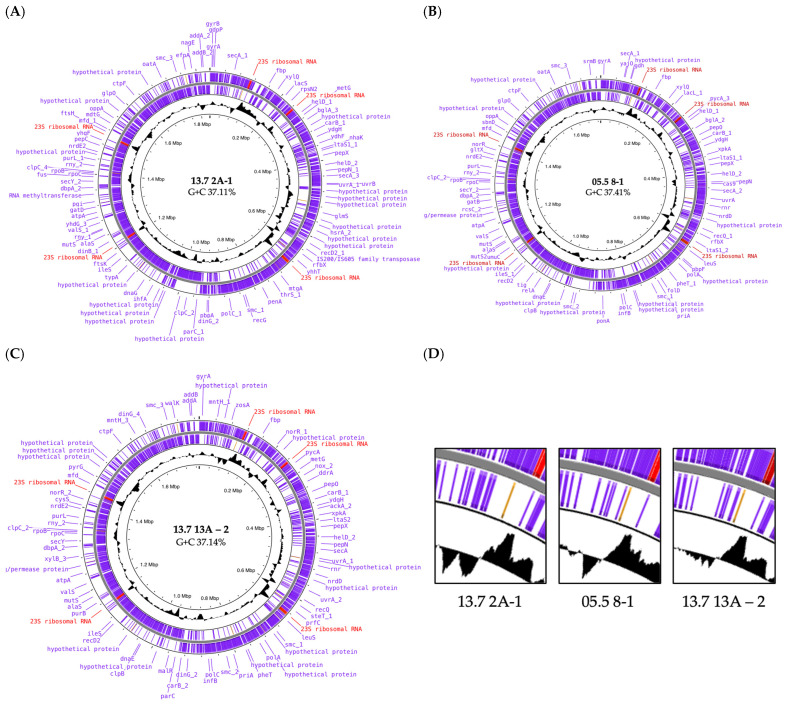
Genomic landscape. (**A**–**C**) show the chromosomal maps of the three indicated *P. pentosaceus* genomes. Genes are in blue, the five 23S rRNA gene loci are labeled red. (**D**) Sections of each genome map around the first 23S rRNA gene locus (red) show the genetic plasticity revealed by the graph of G + C content (black).

**Figure 8 microorganisms-13-02244-f008:**
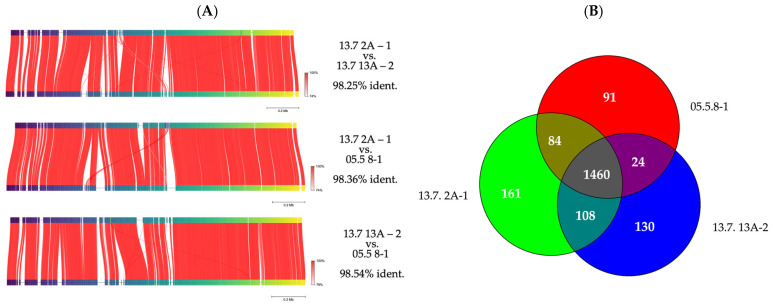
(**A**) FastANI diagrams showing shifts in gene segments or insertion or deletions of binary genome comparisons; (**B**) Venn diagram showing the numbers of genes common to all genomes (core genome), common between two genomes, and the number of singletons.

**Figure 9 microorganisms-13-02244-f009:**
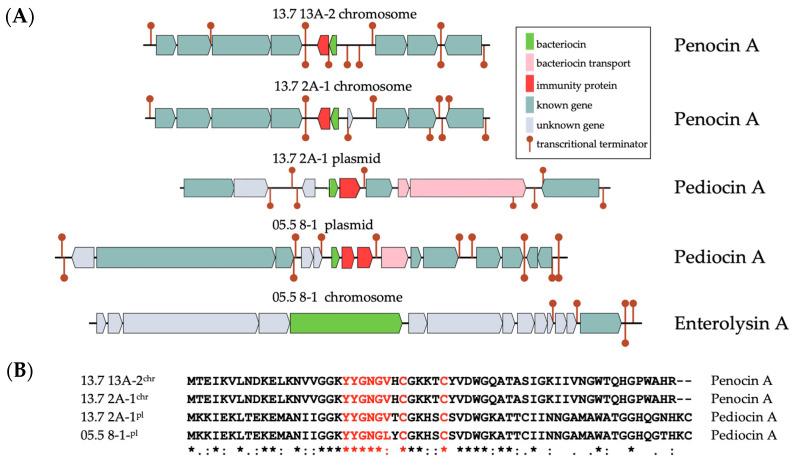
(**A**) Genetic regions of genes involved in bacteriocin metabolism. (**B**) Multiple alignment of bacteriocin protein sequences. The conserved sequence motifs “YGNGV(L) including the two functional cysteines are highlighted in red. ^chr^ chromosomal; ^pl^ plasmidal. The consensus sequence below shows identical (*) and conserved amino acid positions (: and .).

**Table 1 microorganisms-13-02244-t001:** List of *P. pentosaceus* strains.

Strain	Origin	Comment
162-HL-1	farm 1	conventional, milk
64.2-HR-6 ^1^	farm 1	conventional, milk
64.2-HR-4 ^1^	farm 1	conventional, teat canal
92.2-VL-1	farm 1	conventional, foremilk
02.3 3A-1	farm 1	conventional, teat canal
22.4 1A-2	farm 4	conventional, teat canal
22.4 2A-2	farm 4	conventional, teat canal
22.4 4A-2	farm 4	conventional, teat canal
22.4 5A-6	farm 4	conventional, teat canal
22.4 7-1	farm 4	conventional, foremilk
22.4 8A-4	farm 4	conventional, teat canal
22.4 10-2 ^2^	farm 4	conventional, foremilk
22.4 10A-1 ^2^	farm 4	conventional, teat canal
22.4 10A-2 ^2^	farm 4	conventional, teat canal
22.4 11A-4	farm 4	conventional, teat canal
22.4 15-1 ^3^	farm 4	conventional, foremilk
22.4 15A-5 ^3^	farm 4	conventional, teat canal
22.4. 20A-2	farm 4	conventional, teat canal
29.4 13A-2	farm 5	organic, teat canal
29.4 18-2	farm 5	organic, foremilk
29.4 19-1	farm 5	organic, foremilk
05.5 8-1	farm 5	organic, foremilk
13.7 2A-1	farm 13	conventional, teat canal
13.7 17-3	farm 13	conventional, foremilk
13.7 8A-4	farm 13	conventional, teat canal
13.7 10-2 ^4^	farm 13	conventional, foremilk
13.7 10A-6 ^4^	farm 13	conventional, teat canal
13.7 13A-2	farm 13	conventional, teat canal
13.7 18A-3	farm 13	conventional, teat canal
13.7 19A-11	farm 13	conventional, teat canal
13.7 22 A-4 ^5^	farm 13	conventional, teat canal
13.7 22 A-6 ^5^	farm 13	conventional, teat canal

^1, 2, 3, 4, 5^ The isolates were derived from the same bovine from the same quarter of the udder.

**Table 3 microorganisms-13-02244-t003:** Performance ranking of *P. pentosaceus* strains.

Ranking	Strain	No. Indicators Inhibited (z > 0) ^1^	Σ Positive z-Scores ^2^	Mean Positive z-Score ^3^	*p*-ValuePearson ^4^
1	13.7 13A-2	5	2.7508	0.3438	0.1935
2	13.7 2A-1	4	3.0891	0.3861	0.0690
3	05.5 8-1	4	3.0741	0.3843	0.7647
4	22.4 8A-4	4	3.0117	0.3765	0.3096
5	13.7 22A-4	4	2.9012	0.3627	0.0521
6	13.7 18A-3	4	2.8449	0.3556	0.5764
7	29.4 19-1	3	2.8130	0.3516	0.1516

^1^ Count of indicator organisms with above-mean inhibition; ^2^ Cumulative standardized inhibition strength; ^3^ Average standardized inhibition (only z > 0); ^4^ Significance of inhibition vs. pH correlation.

**Table 4 microorganisms-13-02244-t004:** Key data of the genomes.

Strain	13.7 13A-2	05.5 8-1	13.7 2A-1
genome (bp)	1,771,607	1,731,150	1,835,763
predicted genes	1720	1671	1746
plasmids (bp)	12,144; 12,513; 29,159	13,332; 20,517; 21,974; 24,254; 68,459	12,153; 16,367; 22,192; 36,437; 42,648; 44,253
G + C content (%)	37.14	37.41	37.11
*uvrA* ^1^	490,671	464,679	459,969
*polC* ^1^	856,007	806,61	867,588
*secY* ^1^	1,320,181	1,287,919	1,381,749

^1^ Position gene start.

## Data Availability

The original contributions presented in this study are included in the article. Further inquiries can be directed to the corresponding authors. DNA sequences are deposited under the BioProject ID PRJNA1297324.

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
