# Peer review of "Comparative Analyses of *Pediococcus pentosaceus* Strains Isolated from Milk Cattle Reveal New Insights for Screening Food-Protective Cultures"

_microorganisms, 2025, doi:10.3390/microorganisms13102244_

Round 1

Reviewer 1 Report

Comments and Suggestions for Authors

Dear Authors,

In attachment are some comments in order to improve this Manuscript.

The paper presents an overview about the isolation of Pediococcus pentosaceus strains and their potential application as protective culture in foods. The authors performed a significant experimental work that can be relevant for the microbiology community. The overall quality of the study is good, however, I provide some comments that should be addressed before this manuscript could be published. The following suggestions are presented:

Specific points

Line 115: Please correct that the bacteria P. pentosaceus is in italics through the text.

Obtained results should be discussed in more depth.

Line 342: Highlight the mode of action P. pentosaceus against foodborne pathogenic bacteria, its mechanisms.

How environmental factors affect P. pentosaceus action against pathogens?

Try to conclude with a general statement of including future and innovative trends to keep working with the obtained data what will open up new insights for application of P. pentosaceus against the pathogenic bacteria.

Reviewer 2 Report

Comments and Suggestions for Authors

Thank you for the opportunity to participate in the review of the manuscript entitled “A New Strategy to Select Pediococcus pentosaceus Strains for the Application of Protective Cultures in Foods and Care Products”.

The manuscript tells about new strains of Pediococcus pentosaceus and their properties.

Title. “New selection strategy”. Please explain what is new in this selection? These are well-known and used for years studies. I suggest changing the title to something more appropriate to the topic. For example, “New strains of Pediococcus pentosaceus for use as protective cultures in food and personal care products.”

The manuscript has a typical layout for a research paper. The introduction is very short and in my opinion does not introduce the topic well. First of all, it lacks the aim of the work. The introduction does not mention anything about the microbiota of fermented meat products, where Pediococcus occurs very often. Please complete it. The literature cited in the introduction is also not very new. The literature should be supplemented with new items, preferably from recent years (2022-2024). The Material subsection should be supplemented, in accordance with the comments below. The methodology is generally well described, the manufacturers of microbiological media and chemical reagents should be supplemented. The method of presenting the results is correct and very interesting. The authors used new methods of data analysis and used new presentation techniques for this purpose. I have no major comments on the Discussion. The conclusions could be more extensive (they should refer more to the purpose of the work and supplement them). The photos that are in the Supplementary Materials are unacceptable.

From technical matters. In general, the passive voice form should be used in scientific manuscripts. "Something was done ...", not "We did ...". I suggest that the authors familiarize themselves with the way of writing microbiological nomenclature to avoid many corrections in the future.

After a thorough analysis of the text, the reviewer states that the manuscript is not well-written and requires a lot of corrections. The manuscript is suitable for the journal Microorganisms, but requires significant editing.

Here are the reviewer's detailed comments to be added to the text:

Abstract. Please define and include in the abstract the purpose of the work.

Line 20, 21, 209, 210, 213. Gram-negative, Gram-positive are capitalized.

Line 29. Keywords. Please use other words than those in the title of the manuscript. This will increase the possibilities of searching for publications in the database.

Line 41, 50. Please fill in the necessary spaces.

Line 51, 115, 124, 126… Pediococcus and P. pentosaceus are italicized.

Line 61-67. Should this paragraph be placed at the end of the introduction? This text fits better in the Conclusions chapter.

The introduction should end with the purpose of the work. Here the purpose is not defined at all!

Line 69. Please provide all manufacturers of microbiological media and chemical reagents (manufacturer, city, country).

Chapter Material - here should also be a list of Pediococcus bacteria. Below in the figures are presented symbols of strains that were not mentioned earlier.

Line 77. What is Bouillon? Do you mean broth?

Table 1. One of the superscripts is assigned to…?

Line 90. 16S rDNA or 16S rRNA?

Line 120. L. monocytogenes is written in italics.

Line 134, 143. in vivo is written in italics

Line 166. pentosaceus is written in lower case

Line 216. from eight to 17 (please standardize)

Figure 4. Figure 4a and 4b should be signed separately.

Figure 6. The graphs are difficult to read.

The supplementary materials need to be improved. Please prepare photos of the appropriate quality and appropriately signed! Such sloppiness is unacceptable.

Reviewer 3 Report

Comments and Suggestions for Authors

General comments:
The manuscript demonstrates substantial research effort in isolating and characterizing Pediococcus pentosaceus strains. The study is scientifically promising and relevant. However, significant revisions are required to improve data presentation, methodology transparency, and clarity of writing before publication.

The discussion section and conclusion section should be improved. The manuscript mentions similarities to prior research but does not critically compare or differentiate its findings. The authors should include a more detailed comparative analysis with previous research.

The conclusion should be rewritten. 

Additionally, the manuscript inconsistently uses "subspecies" and "strains."  Phylogenetic thresholds are not consistently defined. Clarify terminology and adhere to standard microbiological definitions.

Revise the text for clarity and grammatical correctness.

  • Specific comments: 

Line 10: The resulting food products are tasty and safe. - The product's safety depends on many factors, and we cannot state that it is safe just because a GRASS microorganism is used for its production. Please rephrase. 

Line 37: Metschnikoff postulated that the yogurt-producing lac-35 tic acid bacteria (LAB) Lactobacillus bulgaricus and Streptococcus thermophilus exert health-promoting effects and help humans live longer by acting as intestinal residents.  - To claim something like this, you must cite a clinical study (or even several clinical studies) proving that probiotic bacteria help people live longer. We don't need such sensationalism in the scientific papers. 

Line 41-44. Furthermore, traditional spontaneous (wild) fermentation of food has undergone a renaissance in the past years as a food trend that has been,among other things, strongly acclaimed by chefs at Michelin-starred restaurants to present dishes with novel flavor profiles [8,9].  - The cited papers are from 2016 and 2018, how is it a trend? Additionally, is this a relevant information? I suggest deleting this sentence. 

Line 47-48 The latter two are the most relevant families, of which Lactobacillaceae stand out [11]. - How do they stand out?

Line 51 - substitute increasingly with increasing 

Line 52-54 - Rephrase to be more clear. Avoid word famous

Line 54-56 Strains of P. pentosaceus have become attractive for the fermentation of vegetables, sausages, vegetables, fruits, and wine [16].  - Vegetables are repeated twice. Additionally, wine and sausages are not fermented, they are the product of fermentation of grapes and meat. Rephrase. 

Line 56-58: Numerous health-promoting effects have been associated including microphage stimulation, anti-inflammation, anti-cancer, antioxidant, detoxification, and cholesterol-lowering - The literature cited is not appropriate. Once again, when claiming such information, please cite original clinical studies. Try not to be too sensational and include accurate information. The health-promoting effect of probiotic bacteria is not as big as one might expect after reading this sentence. 

Line 114. In Vivo Detection of Antimicrobial Activity
- How was the pH controlled? Measuring antimicrobial activity without controlling for pH can lead to false positives. The inhibition zone may result from acid secretion rather than specific antimicrobial compounds.
- There is no explicit mention of control experiments to ensure the observed inhibition is due to antimicrobial compounds produced by P. pentosaceus. 
- Why did you use pH strips and not pH meter? Using pH strips is imprecise and introduces variability.  

Line 166 - Isolation of Pediococcus Pentosaceus Strains from the Cattle Udder
-The authors should cite their own published paper that contains the DNA sequencing information. If not, include supplementary data or detailed descriptions to verify these results.

Line 395-397 All isolates exhibited inhibitory activity against diverse pathogenic or food spoiling bacteria making them applicable in food products as protective cultures.  - This is not proved in this paper, you can not claim that the strains are protective cultures, becuase you did not test them on any food. You can only claim that they have the potential to be furhter investigated as protective agents in foods. Rephrase. 

397-398 Hence, the derived data can be used to select the matching isolate for a specific food product that needs a robust food safety along its shelf-life.   - How can be derived data used to match the isolate to a specific food product? This was not even the aim of this study. Please delete. 

General comments: 

Comments on the Quality of English Language

English should be checked and some sentences should be rewritten for better clarity 

Reviewer 4 Report

Comments and Suggestions for Authors

This work aims to isolate Pediococcus pentosaceus in food environment and determine its effects as employing basic indicator and geneomic method. The results are interesting and would provide novel insights in the field of microbial-microbial interaction, and to enhance the food-safety properties in a non-chemical method. The results were nicely presented and described. The discussion was well organized. This work was suggested for minor revision as addressing these points, as shown below.

1、The authors should illustrate the reasons why choose milk cattle as isolation origin

2、Why not choose Staphylococcus aureus as test strain, which are commonly detected in milk cattle?

3、The test results were not verified in the real word (food material), say, the author should confirm this limitation in the MS.

Round 2

Reviewer 2 Report

Comments and Suggestions for Authors

The manuscript has been revised according to the reviewer's suggestions. It is noted that the authors have put considerable effort into improving the text. The manuscript is ready for publication.